# Numerical investigation of the effect of cohesion and ground friction on snow avalanches flow regimes

**Camille Ligneau** [1,2]*, **Betty Sovilla**[1], **Johan Gaume** [2]

**1** WSL Institute for Snow and Avalanche Research SLF, Davos, Switzerland, **2** Snow and Avalanche Simulation Laboratory SLAB, EPFL, Lausanne, Switzerland

\* camille.ligneau@slf.ch

**Data Availability Statement:** All relevant data are within the paper and its Supporting information files.

**Funding:** CL and BS are funded by WSL's strategic initiative Climate Change Impacts on Alpine Mass

## Abstract

With ongoing global warming, snow avalanche dynamics may change as snow cohesion and friction strongly depend on temperature. In the field, a diversity of avalanche flow regimes has been reported including fast, sheared flows and slow plugs. While the significant role of cohesion and friction has been recognized, it is unclear how these mechanical properties affect avalanche flow regimes. Here, we model granular avalanches on a periodic inclined plane, using the distinct element method to better understand and quantify how inter-particle cohesion and ground friction influences avalanche velocity profiles. The cohesion between particles is modeled through bonds that can subsequently break and form, thus representing fragmentation and aggregation potentials, respectively. The implemented model shows a good ability to reproduce the various flow regimes and transitions as observed in nature: for low cohesion, highly sheared and fast flows are obtained while slow plugs form above a critical cohesion value and for lower ground frictions. Simulated velocity profiles are successfully compared to experimental measurements from the real-scale test site of Vallée de la Sionne in Switzerland. Even though the model represents a strong simplification of the reality, it offers a solid basis for further investigation of relevant processes happening in snow avalanches, such as segregation, erosion and entrainment, with strong impacts on avalanche dynamics research, especially in a climate change context.

## Introduction

Snow avalanches are a common natural hazard in mountainous areas. Nowadays, global warming affects snow cover properties such as cohesion and friction [1] which leads to changes in avalanche activity [2, 3]. In the Alps, the current trend shows an increase in the proportion of wet snow avalanches compared to powder snow avalanches [4, 5]. Indeed, an increase of air temperature as expected in the future will result in an increase of snow temperature, which is known to have important consequences for the snow avalanche dynamics by influencing velocities, impact pressures on structures and run-out distances [6–8].

Movements (CCAMM, ccamm.slf.ch) and the canton of Valais. JG is funded by the Swiss National Science Foundation (Eccellenza project: grant number PCEFP2_181227; SPARK project grant number CRSK-2_195914). The funders had no role in study design, data collection and analysis, decision to publish, or preparation of the manuscript.

**Competing interests:** The authors have declared that no competing interests exist.

The snow in the dense part of an avalanche [9] is often considered either as a continuous material [10, 11], or as a mobile granular assembly [12, 13] flowing over an erodible bed—the snow cover—or the ground. The granules can have a diameter ranging from a few millimeters to more than 1 meter [14] and are made of a multitude of snow grains bonded together. Depending on the context of the avalanche, the granules found in the flow can either result from the fragmentation of bigger granules—typically the released slab—or they can be formed by the aggregation of snow grains together. Steinkogler *et al.* [15] defined two processes of aggregation: dry cohesion and wet cohesion. Dry cohesion occurs when two ice crystals come in contact together under the effect of pressure. At that time, they sinter by creating an ice bond between the two crystals [16]. This process is temperature-dependant [17] and is found to happen faster as the temperature increases. Wet cohesion occurs as the temperature gets close to 0˚C. At that point, the presence of liquid water creates bonds by capillarity between snow particles, increasing snow cohesion. However, Steinkogler *et al.* [15] evidenced that water-saturated snow (i.e. slush snow) exhibits very low strength as water becomes a significant phase of the blend.

During the course of an avalanche, bonds between snow grains are formed and broken according to variations of pressure, snow temperature and impact loads. The balance between these aggregation and fragmentation processes leads to the formation or destruction of granules. In this work, we will refer to these two processes with the term *granulation*.

During the flow, the avalanche temperature can either increase through friction, heat dissipation or by the entrainment of warmer snow along the path [18–20]. This means that the erosion process of the warmer snowpack will be particularly important in the context of climate change. However, the link between snow properties and the resulting dynamics of the flow is still poorly quantified and is now becoming essential to improve current avalanches dynamics models.

The rheological properties of the flow change depending on the basal boundary conditions [21], and on the physical and mechanical properties of the snow [15]. Thus, different flow regimes can develop in an avalanche [22], with different characteristics of velocity, shearing, density, pressure and depth [8]. The fickle nature of the snow mechanical properties even allows to find different flow regimes at the same time in a single avalanche [6, 9, 22]. For example, it is commonly observed that the flow characteristics are different between the front and the tail [6, 23]. Moreover, the static snow cover can be eroded by the flow and partially entrained into the avalanche. If the snow cover properties are significantly different from the flowing snow properties, the mixing can critically affect the rheology of the flow leading to a potential flow regime transition, where velocities and pressures can abruptly change in a short time [6].

Köhler *et al.* [22] have identified and characterized four dense regimes in relation to their velocities, flow heights, snow temperature and stopping mechanisms. The *sliding slab regime* occurs just after the avalanche release, usually in the presence of a dry snowpack ($T < -2$˚C, no liquid water content). The snow slab consists in a rigid and fragile aggregate laying on top of a weak snow layer. Under a natural or artificial load, the weak layer collapses and, if the slope angle is steep enough, the slab starts to slide on this weak layer [24]. It rapidly transforms into another regime as the slab fractures because of the stress heterogeneity induced by velocity and/or local topography variations. From there, if the flow temperature does not raise above -1/-2˚C, we would then observe a *cold dense regime*: a quasi-cohesionless dry granular flow with velocities up to 30 m/s, a depth generally below 3 m and large shearing across the depth. If temperature reaches -1˚C, stronger bonds form [16], leading to the formation of persistent granules [15]. As a result, a so-called *warm shear regime* would emerge, with velocities dropping down to 10–20 m/s as cohesion increases, potentially showing higher flow depths. When

the flow temperature reaches 0˚C, liquid water appears. In this case we can observe a *plug flow regime* made of large persistent wet-granules plowing the snow cover at the flow-front. All these regimes can coexist in a single avalanche or they can change from release to deposition if the properties of the snow changes along the path [6, 22]. In addition to these four flow regimes, it is worth mentioning the *slush flow*, which occurs when water from melting and/or rain accumulates in the snowpack, weakens its shear strength and lowers the basal friction until it eventually releases [25]. This flow is constituted of a mixture of snow and free water, with a liquid water content above 15% and densities up to 1000 kg/m$^3$. Velocities up to only 30 m/s have been observed on the field [26], but chute flow experiments suggest that slush flows are actually faster than their dry counterpart in the same conditions [27].

The present work investigates how the cohesion of the snow and the basal layer properties affects the velocity profiles of the flowing snow. Because snow avalanches can be considered as a granular medium, a numerical model implementing the Distinct Element Method (DEM) [28] is used for this study. Most of the research regarding experimental and numerical granular flows consider cohesionless grains where grains interactions are defined through a viscoelastic frictional behavior [29–31]. Much fewer studies consider flows with cohesive interactions between particles such as electrostatic, capillarity, van der Waals forces or solid bridges. Thus, the rheology of cohesive granular materials still widely remains an open field of research. Even if individual cohesive interactions are well defined in the literature [29], cohesive granular studies often implement cohesion as an additional attractive force that pulls particles together when they are in contact [32, 33] to mimic the effect of water meniscus. Studies implementing solid, breakable bonds between particles, as needed to simulate snow, are scarce. Cuellar *et al.* [34] implemented such model to investigate the erosion of an immersed cohesive geomaterial. Several studies focused on snowpack stability [35–39] used a similar contact model to model the sintering of snow as a solid beam joining two particles when they are in contact. The DEM contact model that we use in this work, is based on the latter.

Snow has a complex mechanical behavior, related to its physical state. It is very challenging to explicitly consider the effect of temperature without empirical considerations [40]. In most applications of DEM related to snow avalanches, the temperature is indirectly accounted for through variations of mechanical properties [15]. Here, we follow this approach and study the influence of snow cohesion and ground friction on avalanche flow regimes and qualitatively interpret the results with respect to temperature variations. In addition, despite the recent increase in computational power, the scale still remains problematic in DEM simulations. Hence, most DEM studies concentrated on avalanche formation [35, 41] and avalanche dynamics [12] consider meso-scale simulation setups, focusing on the physical investigation of relevant processes.

The article is organized as follows. In the section *Materials and methods*, we first present velocity profiles that are typical from snow avalanches, extracted from experimental data. The method used to model flowing snow is then described, along with the simulation set-up. Next, details about the averaging technique and data post-processing is given. In the section *Results*, the effects of cohesion and basal friction on the flow properties are shown. Finally, in the section *Discussion*, we examine the numerical results with respect to experimental observations in order to propose physical mechanisms for the observed flow regimes and transitions.

## Materials and methods

### Typical velocity profiles of snow avalanches

Snow avalanches are complex geophysical mass flows that can exhibit various flow regimes that can spatially and temporally vary. In transitional avalanches, the frontal part usually

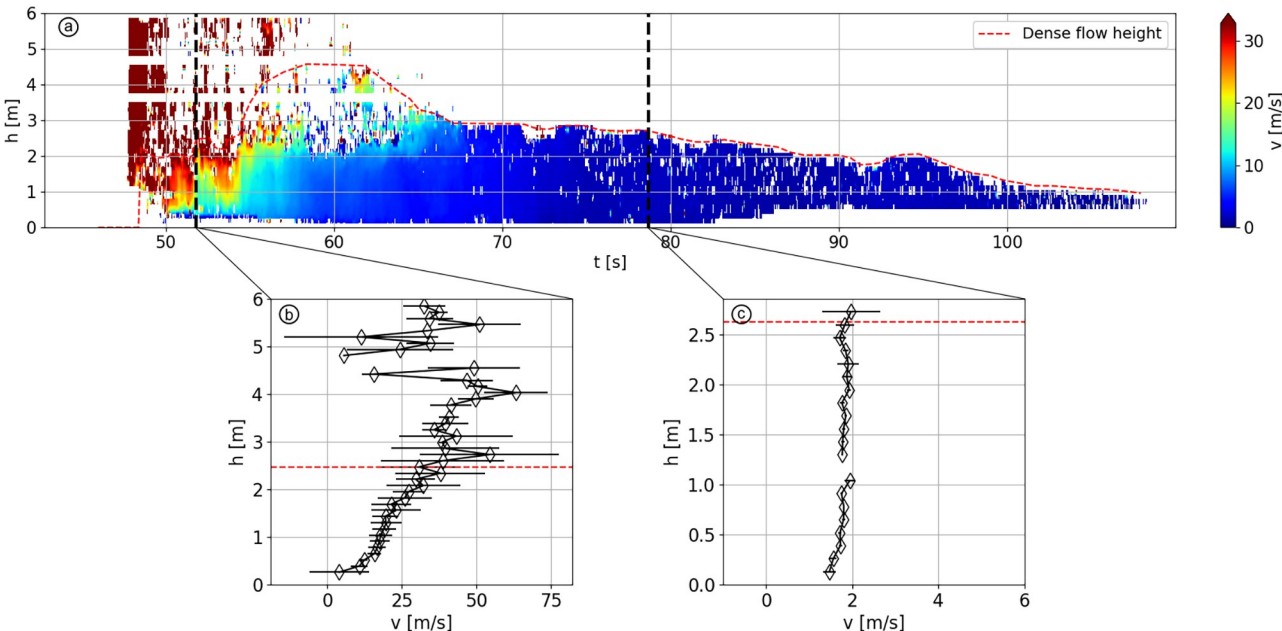

**Fig 1. Velocity data of avalanche #20173032.** (a) Velocity (colorized) and flow depth vs. time. Note that, a flow detachment from the pylon occurred between 55 and 65 sec above 2 meters, resulting in the decreased local quality of the measurements, as shown by the scattered data, below the dashed red line. (b) Velocity profile extracted at the avalanche front ($51.8 < t < 52.3$). (c) Velocity profile extracted at the avalanche tail ($78.7 < t < 79.2$). On the three panels the dashed red lines indicates the height of the dense flow. The average air temperature during the event was -2˚C.

exhibits a sheared, fast velocity profile typical of cold, dry snow. At the tail however, the snow has been already under the effect of internal friction so that its mechanical properties are different from the front. Slow and quasi-vertical velocity profiles are often observed in this part of the avalanche.

We present here velocity profiles from experimental data of snow avalanches that will be used for model comparison. This dataset has been gathered at the real-scale experimental test site of Vallée de la Sionne [42, 43] where avalanches velocities, temperatures or impact pressures are measured with great detail along a 20 meters-high pylon. This pylon is located in the avalanche run-out zone, on a slope of 25˚, at an altitude of 1650 m. Velocity measurements are made on the pylon's side using optical sensors which are recording the time-lag of snow reflectivity between a distance of 30 mm [44], with a vertical resolution of 125 mm.

The avalanche #20173032, captured in Vallée de la Sionne on March 8$^{th}$ 2017, depicts a typical flow regime transition. Fig 1(a) shows the velocity and flow height of this avalanche at the location of the pylon. The red dashed line represents the position of the upper boundary of the basal dense layer. Above this line stands either the dilute powder snow cloud ($48 < t < 55$) or air ($t > 55$). Indeed, only the dense part of the flow is of interest in this study as we aim to simulate dense flow regimes. Panels (b) and (c) of Fig 1 show respectively two velocity profiles averaged over a 0.5 seconds time window at the avalanche front and tail, respectively. We can observe that the front (b) exhibits a fast and highly sheared velocity profile, typical of a dry and cold snow (cold dense regime in Köhler *et al.* [22]), whereas the tail (c) shows a slow and shear-less profile, typical of warm and wet snow (warm plug regime in Köhler *et al.* [22]). We choose to highlight here these two flow regimes as they are commonly encountered in snow avalanches.

## Modeling flowing snow with the Discrete Element Method

We model snow avalanches using a 2-dimensional Distinct Element Method (DEM) [28]. It consists in simulating an assembly of undeformable cylindrical particles allowed to interpenetrate each other of a distance $\delta_{ij}$ (Fig 2a). When $\delta_{ij} \geq 0$, the interaction between particles is accounted for through a contact model with tunable properties of elasticity, viscosity, friction and cohesion. In this study, we use the software PFC v5 developed by Itasca which implements this method.

Here, 2D periodic free-surface flows are simulated. For computational cost reasons, the simulated particles do not represent actual snow grains, which typically have a millimeter-size. They rather represent small snow aggregates of a few centimeters that are often observed in the field [8, 14, 15]. Therefore, the mechanical properties of these simulated particles refer to the mechanical properties of these snow aggregates and not to individual snow crystals. We checked that the two-dimensional character of the simulation affects only marginally the results by comparing velocity profiles of 3D and 2D simulations (S1 Fig).

Particles have a mean radius $\bar{r} = 40$ mm with 20% polydispersity ($32 \geq r \geq 48$ mm) to avoid crystallization. Their density is set to 500 kg/m$^3$, which corresponds, when considering porosity, to a bulk density of 400 kg/m$^3$ commonly measured in snow avalanche deposits [14, 15].

Precise data on the relationship between the mechanical properties of flowing snow and temperature is still largely missing in the literature. Therefore, we choose our model to be purely mechanical, so it does not directly consider temperature. However, the effect of temperature on the cohesion of snow is reflected via a tuning of the potentials of particles bonding and un-bonding defined thereafter.

The contact model (Fig 2b) defines the mechanical behavior between two particles in contact. The contact model used in this work is based on the Linear Parallel Bond Model (PBM)

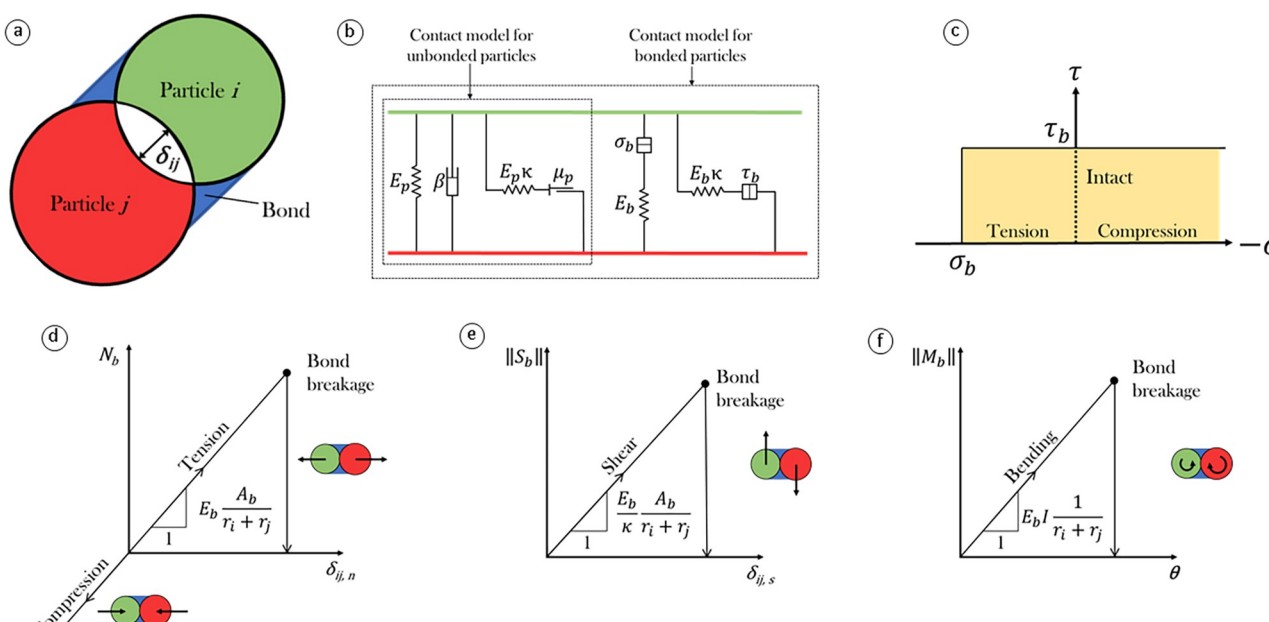

**Fig 2. Parallel bond contact model.** (a) Schematic of two particles $i$ and $j$ being in contact and bonded. (b) Parallel-bond contact model. The values/ranges of denoted values are found in Table 1. (c) Failure envelope of the bond. (d) Bond normal force as a function of normal interpenetration. (e) Bond shear force as a function of tangential interpenetration. (f) Bond bending moment as a function of bending rotation.

**Table 1. Parameters used in the simulations.**

| | Parameter | Symbol | Value / Range | Unit |
|---|---|---|---|---|
| Particles | Mean radius | $\bar{r}$ | 0.04 | m |
| | Polydispersity | | 20 | % |
| | Density | $\rho$ | 500 | kg/m$^3$ |
| | Young Modulus | $E_p$ | $10^6$ | Pa |
| | Friction | $\mu_p$ | 0.5 | - |
| | Damping ratio | $\beta$ | 0.5 | - |
| | Normal-to-shear stiffness ratio | $\kappa$ | 0.5 | - |
| Bonds | Bond creation force | $F_a$ | $10^1$—$10^4$ | N |
| | Young Modulus | $E_b$ | $10^6$ | Pa |
| | Tensile strength | $\sigma_b$ | $10^2$—$10^5$ | Pa |
| | Shear strength | $\tau_b$ | $\sigma_b/2$ | Pa |
| Ground | Friction | $\mu_g$ | 0—0.6 | - |
| | Roughness | $\bar{r}_g$ | $\bar{r}$ | m |

of PFC [45]. However, the latter does not implement the aggregation process, which is described later in this section. This addition to the model was originally performed by Kyburz *et al.* [12] and Steinkogler *et al.* [15]. If particles are not bonded, they interact with a visco-elastic, frictional behavior that allows a relative rotation of the particles, and slip if a Coulomb limit is reached on the shear force. The cohesion is implemented by allowing particles to bond and un-bond under specific conditions. A bond is formed between two particles in contact if the contact force $F_c$ is above the bond formation force $F_a$, which is manually set in the contact model. The contact force between two particles $i$ and $j$ is defined as:

$$F_c = \sqrt{N_{ij}^2 + S_{ij}^2} \tag{1}$$

where $N_{ij}$ and $S_{ij}$ are respectively the normal and shear forces at the contact. The mechanical behavior of the bond (Fig 2b–2f) consists of a purely elastic beam joining the two particles. The bond has dimensions $r_b\, l_{ij}$ where $r_b = min[2r_i; 2r_j]$ is the diameter of the smallest particle in contact and $l_{ij}$ is the distance between the two particles' centers. Thus, two possible interplays are possible between two particles: without or with the presence of a bond. The contact model provides a different behavior depending of this condition (Fig 2b). When a bond is present between two particles, the bond components are active and act in parallel with the unbonded particles components. The bond can break either in tension or in shear if the load exceeds, respectively, the tensile strength $\sigma_b$ or the shear strength $\tau_b$. A new bond is formed every time the condition $F_c > F_a$ is met.

A value of $E = 10^6$ Pa is chosen for the Young's modulus which is in the lower range of reported values for snow [46] in order to improve computational efficiency. In addition, this value allows to keep the interpenetration distance $\delta_{ij}$ small enough to stay in the rigid grain limit [47]. The damping ratio $\beta$ is set to 0.5 as it is assumed that the macroscopic behavior of the particles stays independent of this parameter for $0.1 \leq \beta \leq 0.9$, as it has been shown in previous studies [15, 35, 47]. The friction coefficient of the particles $\mu_p$ is always set to 0.5 which is a typical standard friction value for snow [17, 48]. The values of the tensile strength $\sigma_b$ and shear strength $\tau_b$ are in the range of measurements of Mellor [49] who finds values of 0—0.5 MPa for densities up to 600 kg/m$^3$.

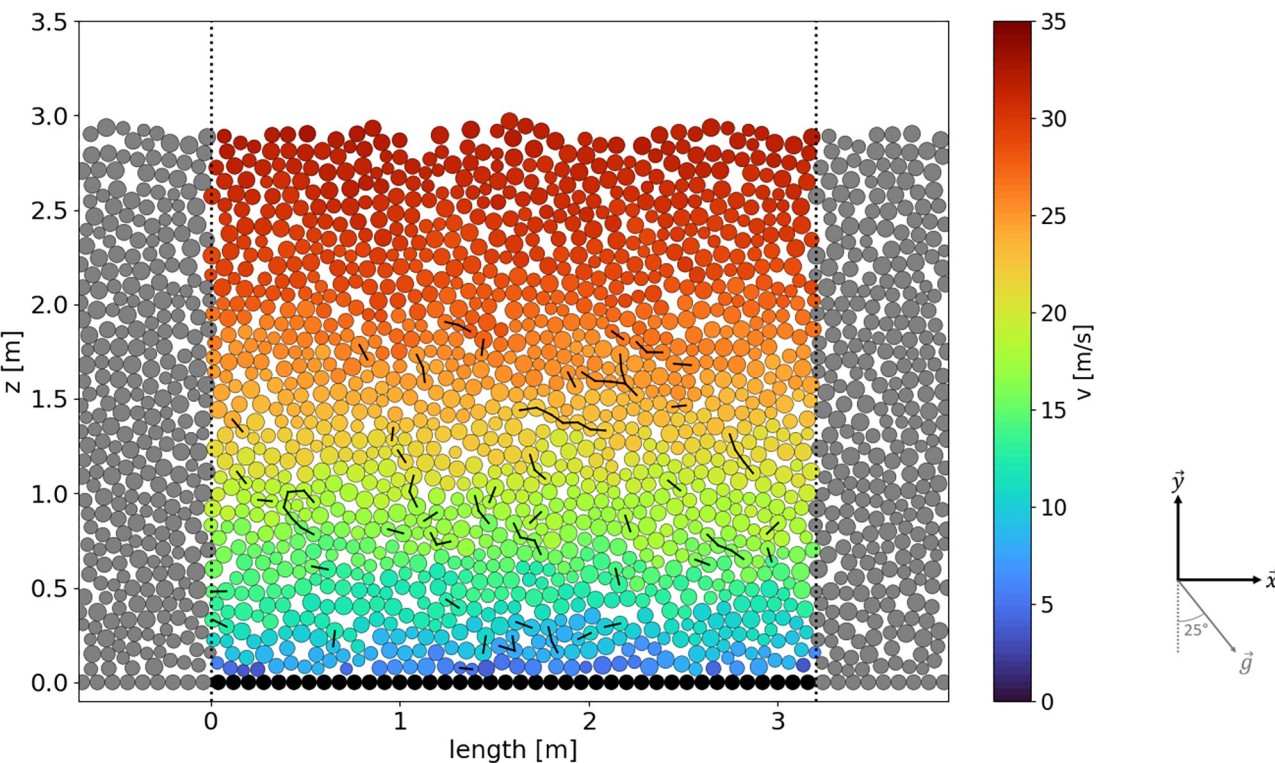

**Fig 3. Setup used for the simulations.** The dotted lines mark the periodic boundaries of the domain. Grey particles are simply a replica from the domain's particle and are only shown for illustration. The velocity of particles (colorized) and the bonds between particles (black segments) are shown as an example.

## Model setup and simulation procedure

We model snow flowing on a slope with a two-dimensional DEM in order to reproduce different flow regime scenarios. DEM allows to get information on every single particle, such as velocity, contact force and bond status. By tuning particle and bond parameters, in particular the values $\sigma_b$ and $F_a$ related to cohesion, we can reproduce different flow regimes.

The setup's domain (Fig 3) has a length $L = 3.2$ m and no height limit. The length is chosen long enough to ensure that the results are size-independent over the $\vec{x}$ axis. We verified that the results are not significantly affected if larger values of $L$ are used. Gravity is set with an angle of 25° from the $\vec{y}$ axis, representing the slope inclination around the pylon in Vallée de la Sionne. Periodic boundary conditions are applied along the $\vec{x}$ axis, so that all the particles leaving the domain on the right-hand side will reappear on the left-hand side (i.e. on top of the slope) with the same mechanical properties. That way, it is possible to model continuous flows with constant flow heights.

The ground layer is made of particles of radius $r_g$ as shown in Fig 3, giving the ground a roughness. Except from the friction value, the ground particles have the same properties than the ones in the flow, with the exception that they cannot bond with other particles. In order to keep a fixed ground, the ground particles are also blocked in all directions and rotation.

At the beginning of a simulation, particles are randomly placed in an area of length $L$, the length of the domain, and height $H = 2.5$ m until a volume fraction of 0.85 is reached, which gives approximately 1300 particles. The initial value of the volume fraction corresponds to a tight packing of mono-disperse particles in 2D, but decreases as soon as the particles are in movement due to dilatancy. A typical value for the volume fraction during a simulation is 0.8,

so the bulk density of the medium is around 400 kg/m$^3$ since the particle density is 500 kg/m$^3$. Unless specified otherwise, simulations start with an non-bonded state, even if some can be naturally present if the conditions described in section *Modeling flowing snow with the Distinct Element Method* are initially met ($F_c > F_a$).

The values of $\sigma_b$ and $F_a$ are set at the beginning of the simulation and kept constant, except for transitional flows where they are gradually changed during the simulation. In this case, we wait that the flow reaches a first steady state before applying the changes.

The slope angle is set to 25° and is therefore very close to the angle of friction of the particles ($\approx$ 26.6° for $\mu = 0.5$). For this reason, depending on the value of the ground friction, the granular medium may need an initial velocity $V_{ini}$ to start flowing. This trigger is implemented by applying an initial velocity on the particles on the first timestep. The flow then develops and, in time, the kinetic energy of the system reaches a steady state as the frictional forces and dissipation due to collisions compensate the gravitational forces.

## Averaging and coarse-graining

The analysis of the simulations is made while the flow is in a steady state, except for flow transitions. We assume that the steady state is attained when the kinetic energy of the particles reaches a steady value. Starting from this instant, the positions, velocities and contact forces of the particles are recorded every 0.1 seconds for a time of 10 seconds. The velocity and stress profiles are then computed from this data (S2 Appendix). To do so, a coarse-graining method is used [50], allowing to pass from the Lagrangian description of the particles given by the DEM, to an Eulerian description of the granular flow.

First, the domain is discretized with a grid having vertical and horizontal resolution of $2\bar{r}$. The values of velocity and stress are computed on each point of this grid by weighting Lagrangian values within a circle of diameter $h$ around the measurement point with the following Lucy weight function:

$$\mathcal{W}(\mathbf{r}) = \frac{5}{\pi h^2} * \left(1 - 6\frac{r^2}{h^2} + 8\frac{r^3}{h^3} - 3\frac{r^4}{h^4}\right) \quad \text{if} \quad 0 \leq r \leq h \tag{2}$$

with $r$ being the distance between the grid point and the microscopic value coordinates (particle center for the velocity, contact point for contact forces). For this study, a value of $h = 1.2\bar{r}$ is chosen in accordance with the recommendation of Weinhart *et al.* [50] to have smooth macroscopic profiles.

From the coarse-grained values of velocity and stress computed over a grid of dimensions ($x \rightarrow 2\bar{r}L$; $z \rightarrow 2\bar{r}H$; $t \rightarrow 100$) we average these values over the $x$-axis and the $t$-axis, so that the profiles have a vertical resolution of $2\bar{r}$. By using this method, we make the hypothesis that physical parameters vary across $\vec{z}$ but are constant along $\vec{x}$ and stable in time. For more details about the calculation method of velocities and stresses, the reader can refer to Weinhart *et al.* [50] and the references therein.

## Fragmentation and aggregation numbers

In order to characterize the simulated flows, the parameters of the contact model are linked with the properties of the flowing snow. Two dimensionless numbers are used, that were introduced by Steinkogler *et al.* [15] when studying the granulation of snow in a rotating tumbler. The aggregation number $\eta_a$ characterizes the potential of bond formation in the system, while the fragmentation number $\eta_f$ characterizes the potential of bond breakage.

The definition of $\eta_f$ presented here differs from Steinkogler *et al.* [15], as they consider that the main source of bond breakage is particle collisions. Here however, the main source of

bonds formation and breakage is the stress induced both by hydrostatic pressure and particles motion, as the simulated flows mostly stay in the frictional regime, except close to the free surface in cases with low cohesion. Hence, we do not account for the collisional contribution in the definition of $\eta_f$. We define the aggregation and fragmentation numbers as:

$$\eta_a = \frac{\langle\sigma\rangle \bar{A}_b}{F_a}, \qquad (3)$$

and

$$\eta_f = \frac{\langle\sigma\rangle}{\sigma_b}, \qquad (4)$$

with $\langle\sigma\rangle$ being the depth-averaged stress in the flow at its steady state, and $\bar{A}_b$ the mean surface of a bond's cross section. Here, particles are always cylinders of mean radius $\bar{r}$ and have unitary length, so $\bar{A}_b = 2\bar{r}$. The aggregation number $\eta_a$ is controlled by the value of $F_a$, the bond formation force in the contact model, while the fragmentation number $\eta_f$ is controlled by the strength of the bond $\sigma_b$. $F_a$ and $\sigma_b$ are tuned in every simulation to reach targeted values of $\eta_f$ and $\eta_a$, depending on the desired scenario. In the following sections, we will use the value $\sigma_a = \frac{F_a}{2\bar{r}}$ to conveniently compare stress values. Also, the link can already be made here between $\eta_f$, $\eta_a$ and cohesion: for low values of aggregation and high values of fragmentation we will talk about a low cohesion, and the other way around for high cohesion.

## Results

In a snow avalanche, the increase of snow temperature above -2˚C triggers an increase in snow cohesion, itself triggering the formation of granules—or aggregates—which affects the flow dynamic [15]. In this section, we reproduce this mechanism with DEM by tuning the formation/breakage of bonds between the discrete particles. The fragmentation and aggregation numbers permit to set the potential of formation and breakage of the bonds. The values of the parameters $\sigma_b$ and $F_a$ are adjusted to reach the desired values of $\eta_f$ and $\eta_a$, respectively. We explore thereafter the effect of this tuned cohesion and other parameters on velocity profiles. The range $0.1 \leq \eta_f \leq 10$ is explored as it is corresponding to realistic strength values for snow. The same range is explored for $\eta_a$, despite the lack of experimental values for the bond aggregation force. Indeed, this rather large range allows us to investigate the influence of $\eta_a$ in a parametric approach.

Fig 4 shows the velocity profiles for simulations in the ($\eta_a$; $\eta_f$) domain. For high fragmentation and low aggregation values (upper left corner, low cohesion), sheared profiles are observed since we get closer to the cohesionless case. On the other hand, for lower fragmentation and higher aggregation (lower right corner, high cohesion), the cohesion jams the particles together, causing the flow to stop.

First, the effects of fragmentation and aggregation potentials on sheared velocity profiles are examined. Then, we investigate the conditions required to get a plug flow as it can be observed in snow avalanches, for example the one displayed in Fig 1c. Finally, we simulate a flow transition from a quasi-cohesionless flow to a plug flow and compare it with the experimental data of avalanche #20173032 presented in section *Typical velocity profiles of snow avalanches*.

Throughout this section, we present seven scenarios with different fragmentation and aggregation number and initial conditions. Table 2 summarizes the applied cohesion parameters, as well the observed profile shape and predicated snow types and flow regimes.

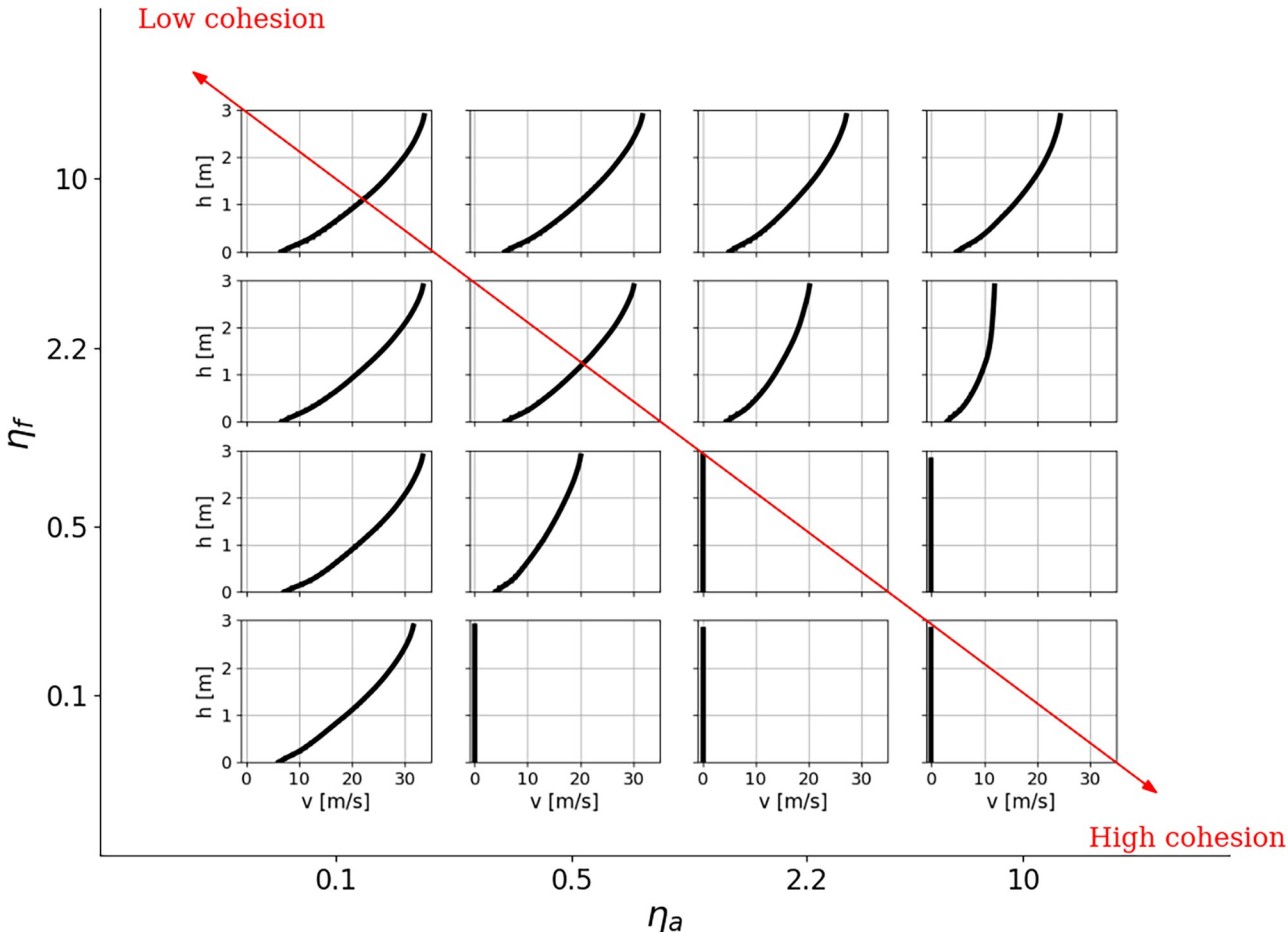

**Fig 4. Velocity profiles for different combinations of aggregation number $\eta_a$ and fragmentation number $\eta_f$.** For these simulations, $V_{ini}$ = 1 m/s and $\mu_g$ = 0.5.

## Sheared velocity profiles of cohesive flows

The scenario 1 aims to simulate the flow of a cohesionless snow. Therefore, we set $\eta_f$ = 10 and $\eta_a$ = 0.1 to reach a high fragmentation with a low aggregation potential. As expected, the resulting velocity profile (Fig 5a) shows a Bagnold-type profile with a shear rate being maximum close to the bottom of the flow and decreasing towards the flow surface. In this example, the slip velocity is roughly 7 m/s while the surface velocity is around 33 m/s. Fig 5b shows that

**Table 2. Parameters and observations on the simulated flow regime scenarios.**

| Scenario | Fig | $\eta_f$ | $\eta_a$ | Other applied parameters | Predicated snow type | Profile shape | Predicated flow regime |
|---|---|---|---|---|---|---|---|
| 1 | 5a–5c | 10 | 0.1 | - | Cold loose | Bagnold | Cold dense |
| 2 | 5d–5f | 10 | 10 | - | Slush | Bagnold | Slush flow |
| 3 | 5g–5i | 0.9 | 2.2 | - | Warm | Bi-viscous | Warm shear |
| 4 | 6a–6c | 0.1 | 0.1 | - | Cold loose | Bagnold | Cold dense |
| 5 | 6d–6f | 0.1 | 0.1 | Initial cohesion | Cold slab | Vertical | Sliding slab |
| 6 | 6g–6i | 0.1 | 0.1 | Initial cohesion and shear rate | Fractured cold slab | Bagnold | Cold dense |
| 7 | 8 | 0.1 | 10 | Initial velocity, lower $\mu_g$ | Warm | Vertical | Warm plug |

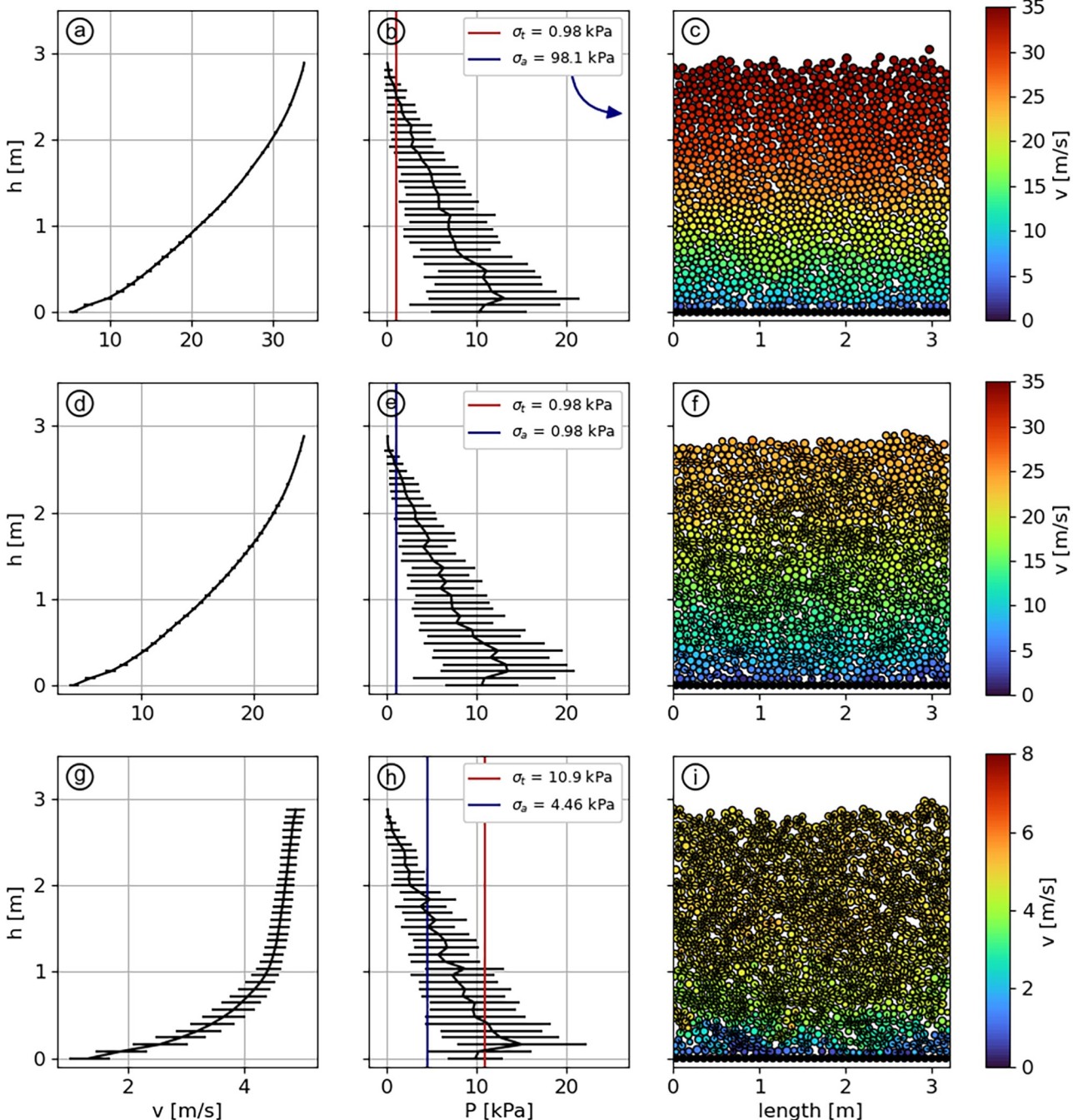

**Fig 5.** Velocity profiles (a,d,g), pressure profiles (b,e,h) and particles visualisations (c,f,i) for scenarios 1 (first row, $\eta_f = 10$, $\eta_a = 0.1$), 2 (second row, $\eta_f = \eta_a = 10$) and 3 (third row, $\eta_f = 0.9$, $\eta_a = 2.2$). Red and blue lines show $\sigma_b$ and $\sigma_a$, respectively, for each scenario. Note that both lines are superposed in panel (e). The error bars display the standard deviation calculated from the averaging over $t$ and $\vec{x}$.

$\sigma_a > P$ for the whole depth, which means that the stress is not high enough to create bonds. Additionally, $\sigma_b$ is lower than the pressure, with the only exception at the flow surface, which implies that even if a bond is formed, it would instantly break under the stress. This behavior is verified with Fig 5c where we can see that no bonds are present within the material.

The scenario 2 aims to simulate the flow of a slush snow. Steinkogler *et al.* [15] observed that this type of snow is simulated with high aggregation and fragmentation values, therefore we set $\eta_f = \eta_a = 10$. In this case, bonds are easily formed as $\sigma_a < P$. However, as $\sigma_b$ is low, the bonds are also broken without difficulty (Fig 5e). Nevertheless, Fig 5f confirms that many bonds are still present, compared to scenario 1, because of the higher aggregation potential. We clearly see the effect of this behavior reflected on the velocity profile (Fig 5d), which has a similar shape than in the cohesionless case, but with an overall smaller velocity (range 5—25 m/s).

The scenario 3 aims to simulate the flow of a snow having intermediate values of $\eta_a$ and $\eta_f$. This corresponds to warm, cohesive snow in the temperature range -1˚C/0˚C, generating persistent moist granules according to Steinkogler *et al.* [15]. We set $\eta_f = 0.9$ and $\eta_a = 2.2$, corresponding to $0 < \sigma_a < \sigma_b < P_{max}$ (Fig 5h) and observe two distinct regions characterized by different mean shear rates (Fig 5g). Indeed, between 0 and 1 m, the sheared profile ranges from 1.6 to 4.4 m/s, whereas the portion between 1 and 3 m exhibits an almost constant velocity in the range 4.4—4.9 m/s. In the upper part of the flow, the stress is high enough to form bonds, but not high enough to break them. It results in a large number of permanent bonds in the top layer (Fig 5i), making the granular medium move almost as a single block, hence the quasi-vertical velocity profile. However, in the lower part of the flow, $\sigma_b < P$, even if $\sigma_a$ is still lower than $P$, and the bonds can be broken by a sufficient stress. Thus, despite the high number of bonds present in this lower part, we can still observe a sheared velocity profile.

Scenarios 4, 5 and 6 aim to simulate flows characterized by low potentials of aggregation and fragmentation of the particles: $\eta_a = \eta_f = 0.1$. Here, $\sigma_a$ is high enough so that bonds are rarely created. But the high strength of the bonds makes them difficult to break (Fig 6b, 6e and 6h). Because neither of the granulation mechanisms will take over the other as the material flows, we expect the flow regime to be strongly dependent of the initial bonding conditions as bonds that are originally present would persist and affect the flow. For this reason, scenarios 4, 5 and 6 are initialized with different conditions but with similar $\eta_a$ and $\eta_f$. Scenario 4 is initialized like scenarios 1, 2 and 3: no initial bonding and no initial velocity. Scenario 5 is configured with an initially-bonded state where all particles in contact are already bonded before starting the simulation, resulting in a brittle slab prone to fracture if a sufficient stress is applied. Yet, such a configuration is not flowing without an initial disturbance, as some slab's particles are trapped by the ground roughness, immobilizing the whole assembly. An initial velocity of 5 m/s with a vertical velocity profile is then applied to initiate the flow. Finally, scenario 6 aims to represent the same slab as scenario 5, but after its fracturing. To achieve this, an initial bonding is also set before the simulation start, and a sheared velocity profile $V(z) = 10\frac{z}{H}$ is applied, thus initiating fractures within the slab.

In scenario 4, the velocity profile (Fig 6a) is similar to the cohesionless case of scenario 1, with a Bagnold-type profile in the range 6–32 m/s. The only change being the presence of a few bonds within the material (Fig 6c).

In scenario 5, the initial velocity generates a shear in the area of the slab in contact with the ground, where bonds start to break. While the flow develops, more bonds are broken in the same area as they are sheared between the static ground and the gravity-driven solid slab. This results in a bi-layered velocity profile (Fig 6d). Close to the ground, a layer of about 60 cm made of unbonded particles shows a highly sheared profile with a velocity ranging from 6 m/s to 12.5 m/s. On top of this layer, the original slab is still intact (Fig 6f) and slides over the sheared layer with a velocity of 12.5 m/s, constant across the slab's depth.

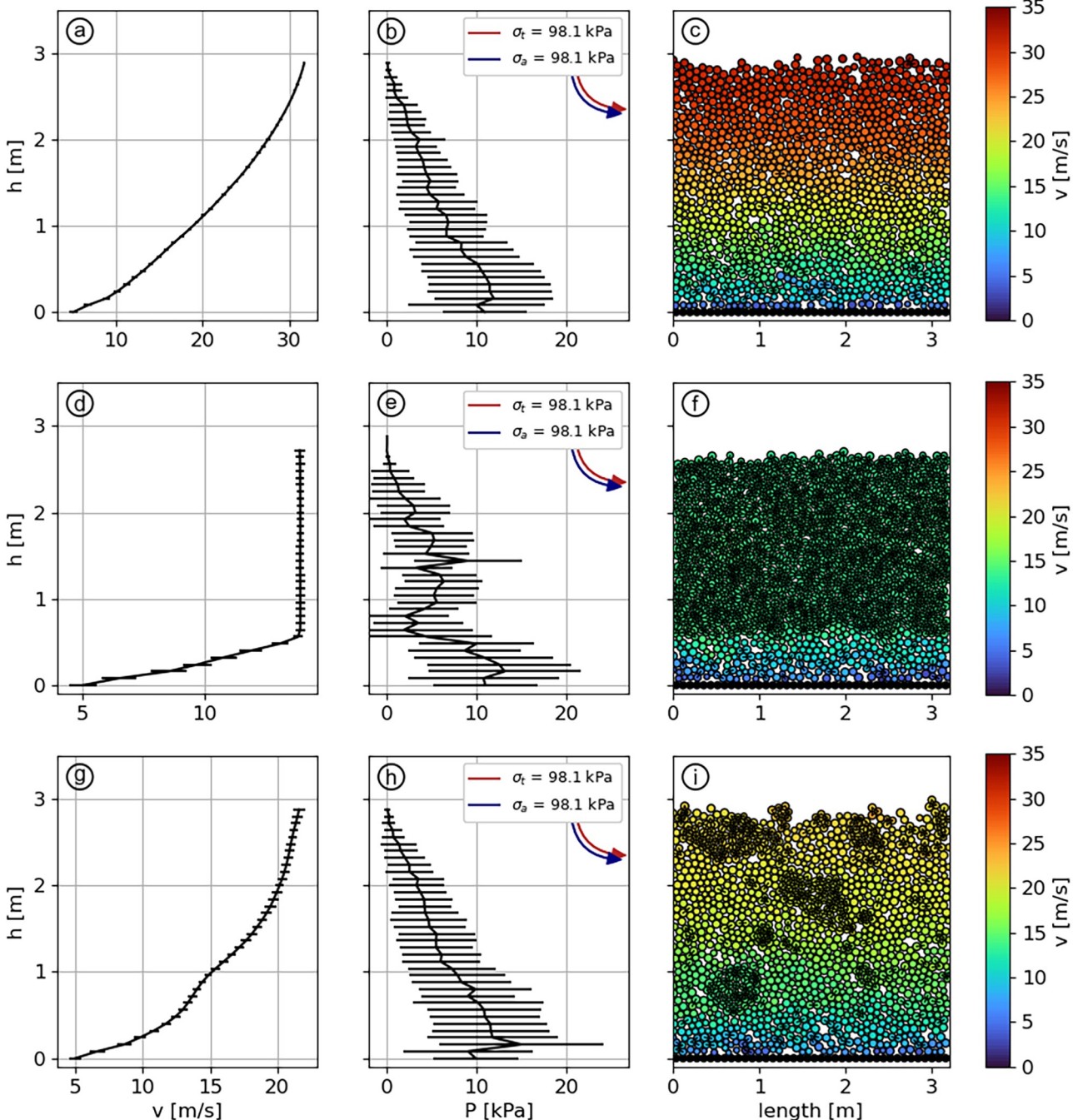

**Fig 6.** Velocity profiles (a,d,g), pressure profiles (b,e,h) and particles visualisations (c,f,i) for scenarios 4 (first row), 5 (second row) and 6 (third row). $\eta_f = \eta_a = 0.1$. Values of $\sigma_b$ and $\sigma_a$ are out of range in panels (b,e,h) but are indicated. The error bars display the standard deviation calculated from the averaging over $t$ and $\vec{x}$.

Finally, scenario 6 exhibits a Bagnold-type velocity profile (Fig 6g), with a velocity of 6 m/s close to the ground and 19 m/s at the surface. The initial sheared velocity profile applied at the simulation's start fractured the slab, breaking it into a mix of fine, cohesionless particles and a few solid granules, the latter being carried by the cohesionless particles (Fig 6i).

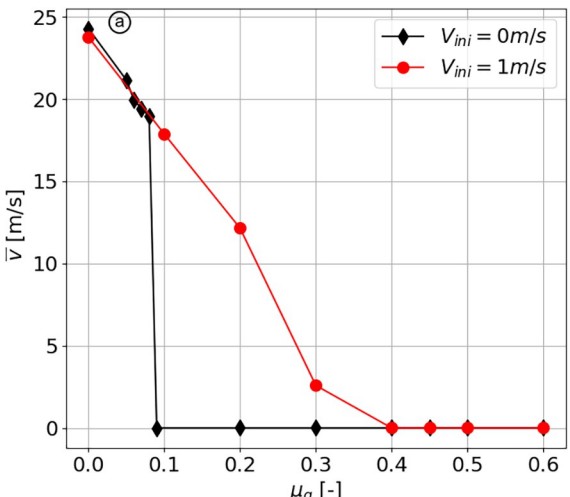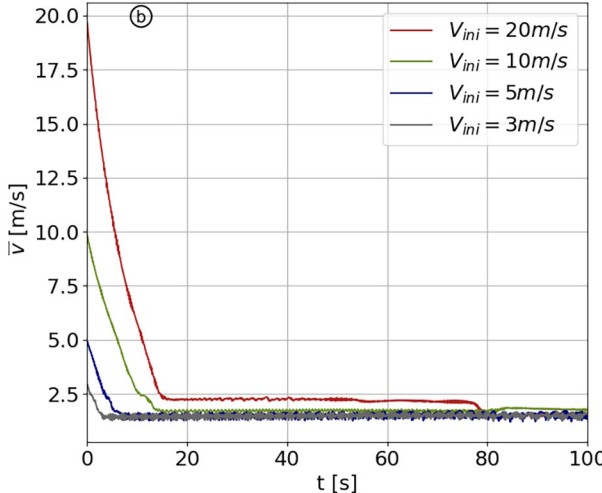

**Fig 7. Effect of ground friction and initial velocity on plug flows.** (a) Mean velocity vs. ground friction after the time necessary for the flow to reach a steady state. (b) Mean velocity vs. time for $V_{ini}$ = 3, 5, 10 and 20 m/s and $\mu_g$ = 0.35. All simulations are run with $\eta_f$ = 0.1 and $\eta_a$ = 10.

## Plug flow regime

In snow avalanches, plug flow regimes, which are characterized by a vertical velocity profile, have velocities in the range 1–5 m/s. This kind of flow regime is typical for wet snow avalanches and usually also arises in dry avalanches undergoing a regime transition, toward the avalanche tail where the snow is warmer and more cohesive (Fig 1c). However, the velocity profiles displayed in the lower right corner of Fig 4 shows that a high cohesion leads to a stop of the flow, whereas actual avalanche plug flows can run for several minutes, even on slopes having very low angles [6]. A high cohesion is therefore not the only necessary parameter for a plug flow regime to occur. This section investigates the required boundary conditions to form a plug flow. We focus here on the effects of ground friction and the initial velocity given to the flow.

In the simulations presented in Fig 4, a plug flow can not occur because snow particles in the lower layer are trapped in between the ground particles as a result of robust bonds and high hydrostatic pressure preventing them to move over the ground roughness. By gradually decreasing the value of ground friction $\mu_g$, without initiating the flow with an initial velocity (Fig 7a in black), we find that the pack of particles start to flow from a ground friction of 0.09. Below this value, the steady state velocity of the flow directly reaches 20 m/s, which is not in accordance with field measurements collected at our observation station in Vallée de la Sionne, where plug flows exhibit typical velocities below 5 m/s. Therefore, the ground friction plays a role in the formation of plug flows, but is not, together with high cohesion, the only required parameter.

To prevent the previously described behaviour, we impose an initial value of velocity of 1 m/s (Fig 7a in red) to initiate the plug flow. For $0.01 \geq \mu_g \geq 0.4$, the initial velocity of the flow directly impacts its flowing state. Above this range the plug flow stops anyway, whereas below this range the flow velocity is similar than without the initial velocity. The magnitude of $V_{ini}$ is shown to be insignificant for the final velocity of the flow (Fig 7b), as for values of 3, 5, 10 and 20 m/s, all flows decelerate to $\approx$ 2 m/s. The scenario 7, shown in Fig 8, aims to reproduce a plug flow by implementing the aforementioned range of parameters, in this case with $\mu_g$ = 0.35 and $V_{ini}$ = 10 m/s.

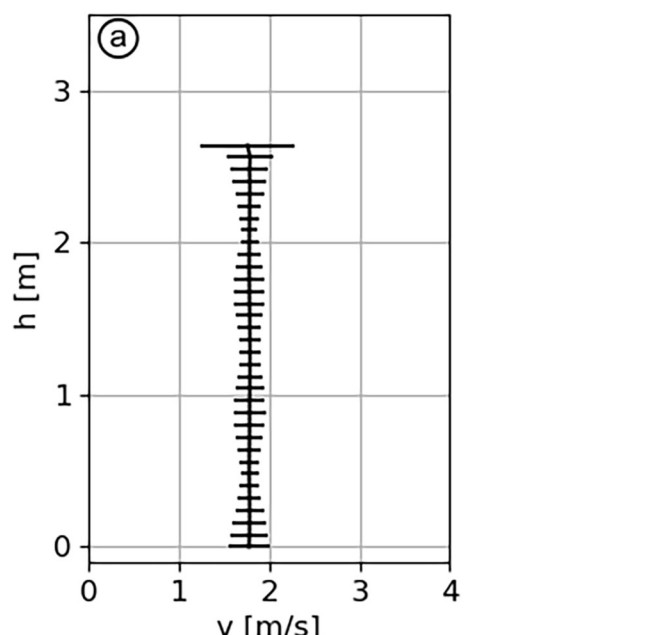
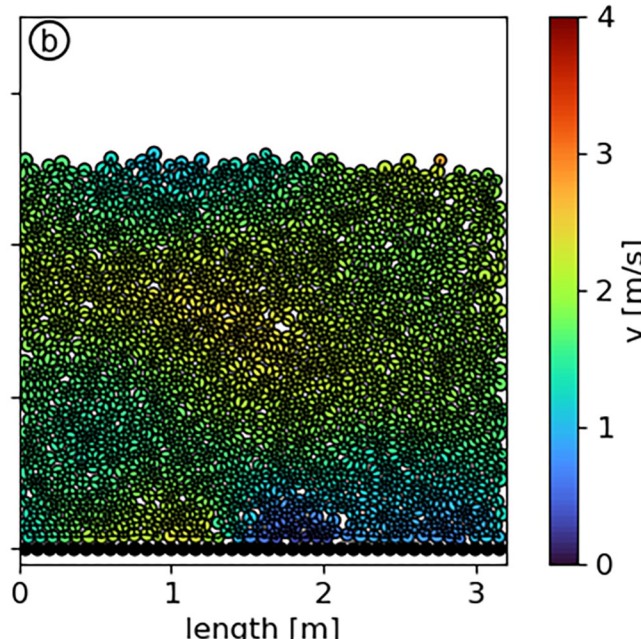

**Fig 8.** (a) Velocity profile, (b) particles vizualisation for scenario 7. $\eta_f = 0.1$, $\eta_a = 10$, $\mu_g = 0.35$ and $V_{ini} = 10$ m/s. The error bars display the standard deviation calculated from the averaging over $t$ and $\vec{x}$.

## Comparison of experiments and simulations

Velocity profiles from the simulations and from field measurements are compared in Fig 9. The data are extracted from the measurements of the avalanche #20173032, presented in Fig 1, and represent two extreme scenarios: high velocity and shear versus low velocity and plugged profile. In the several simulated scenarios presented in sections *Sheared velocity profiles of cohesive flows* and *Plug flow*, two of them, 1 and 7, exhibit similar characteristics both in terms of cohesive properties and simulated velocity profile.

The first experimental profile is captured at the avalanche front (Fig 9a). The profile is the same than the one showed in Fig 1b, except that the speed of the turbulent cloud running over the dense part is cropped, as it is not within the scope of our research. The corresponding simulation is the scenario 1, which has a high potential of bond breakage and a low potential of bond formation. The second experimental profile is captured at the avalanche tail (Fig 9b). It is similar to the one of scenario 7, which aims to simulate a plug flow.

The flow regime transition between the front (Fig 9a) and the tail (Fig 9b), occurs smoothly, lasting for approximately 40 seconds (Fig 10b in blue). To reproduce this transition with the model, we start from the cohesionless flow of scenario 1, and gradually change the values of $\sigma_b$, $F_a$ and $\mu_g$ to the values used for scenario 7. As we aim to reproduce the transition between these specific velocity profiles without considering the intermediate dynamic, these values are linearly changed over a period of 20 seconds, to reduce the computational time (Fig 10c).

## Discussion

The results of the present study shows that snow avalanche flow regimes and transitions can be modeled by a cohesive granular medium flowing down an inclined plane. The contact law of interacting particles can be tuned to mimic the mechanical properties of snow, giving rise to

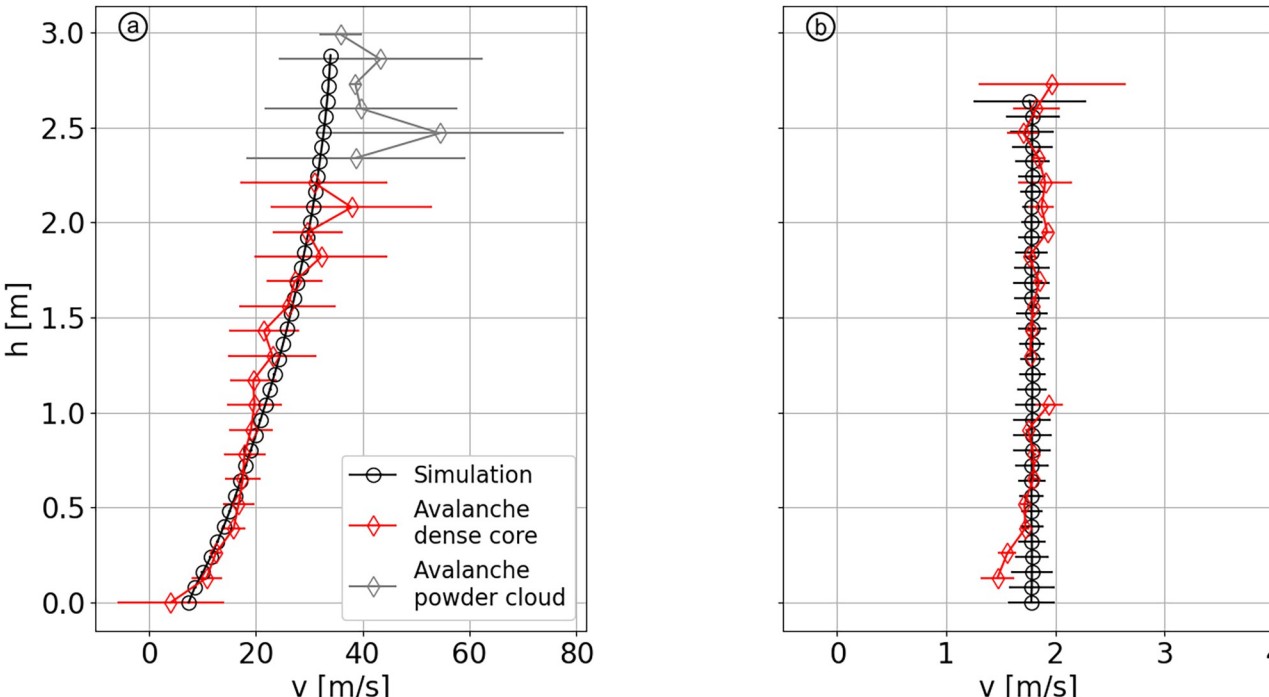

**Fig 9. Comparison of experimental (red) and simulated (black) velocity profiles.** (a) Profile from Fig 1b compared with scenario 1. (b) Profile from Fig 1c compared with scenario 7.

various flow regimes. In this section, we first discuss the ability of the contact model to simulate actual flowing snow. Then, we discuss the effect of snow properties and ground conditions over the flow regimes. Finally, we list the limits of our setup and give recommendations for possible improvement of the simulations.

## Modeling flowing snow with DEM

The contact model used in this work implements the cohesion between particles by modeling solid bonds attached to them. Their existence is set through two distinct processes: the potentials of bond creation (aggregation) and breakage (fragmentation). In actual snow, these processes are respectively controlled by the formation rate and the strength of the bonds. In the literature, the formation rate of bonds has been found to be temperature dependant for wet snow [15] as a higher temperature generates more water, hence more bonds, in the limit of water saturation. For dry snow however, it is the grain size that affects the most the sintering rate, while temperature plays a less significant role [51, 52]. Regarding the bonds strength, it is found to increase for dry snow as the temperature gets close to 0°C [16], but is constant for wet snow as liquid menisci stay at 0°C. The formation rate and the bonds strength are thus differently impacted by various physical parameters such as the presence of water, temperature or grain size. For this reason, we decided to separate the processes of aggregation and fragmentation in the model to control them independently.

In snow, the bonds themselves are either solid ice bridges in the case of dry snow cohesion or liquid water menisci for wet snow cohesion. Our contact model seems then properly adapted to simulate dry snow particles, but the modeling of liquid bridges is debatable due to the purely elastic behavior of the simulated bonds. In literature, most of the models used for

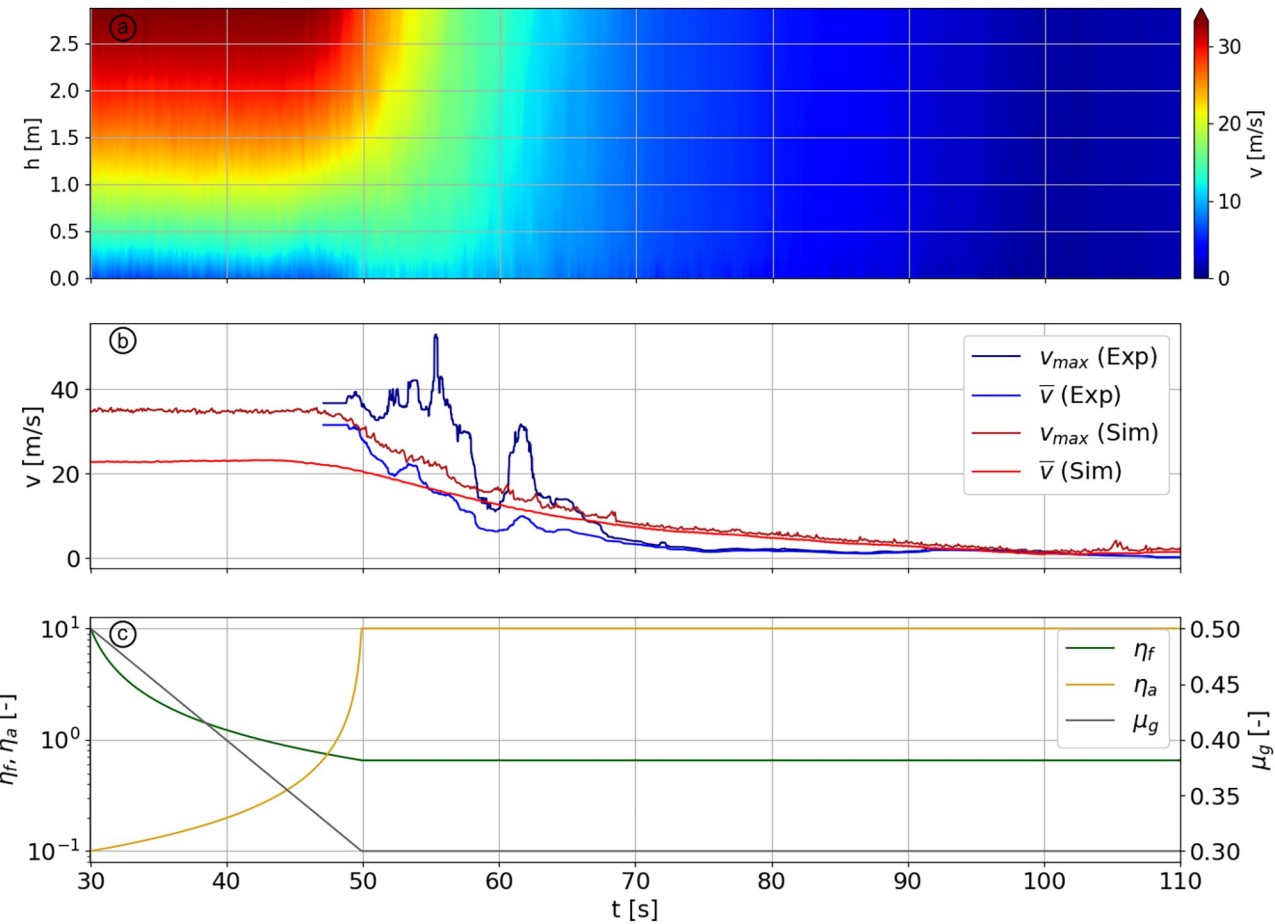

**Fig 10.** (a) Velocity (colorized) vs. height during a DEM simulation of transitional flow regime. (b) Maximum and average velocity of avalanche #20173032 vs. time in dark blue and blue, respectively. The figure also shows the maximum and average velocity of the DEM simulations vs. time in dark red and red, respectively. (c) Evolution of parameters $\eta_f$, $\eta_a$ and $\mu_g$ vs. time in the simulation.

cohesive granular flows usually regulate cohesion through an additional force acting between particles as soon as they are in contact [32, 33, 53]. This force is attractive until the interpenetration of the particles reach a given value and becomes repulsive from a critical overlap, resulting in a behavior coming quite close to the reality of liquid bonds. Despite the solid nature of our bond model, the existence of an aggregation force threshold mimics the attractive potential used in common cohesive contact models, which also justifies its use for wet snow and allows to model transitions from dry (high $\eta_f$, low $\eta_a$) to wet (low to average $\eta_f$, average to high $\eta_a$), and even slush (high $\eta_f$ and $\eta_a$). Such diversity in the behavior of flowing snow could not be modeled with cohesive contact model that do not isolate the processes of bond formation and bond breakage.

In an ideal simulation, the cohesion and friction of particles would be directly linked to temperature. However, the relationship between the mechanical properties of flowing snow and temperature is still missing. Therefore, the temperature is not directly considered in our purely mechanical model, but its impact on the cohesion of snow is reflected via the tuning of the fragmentation and aggregation potentials. In the next section, we discuss this relationship in a qualitative way.

### Effect of snow properties and ground conditions on avalanche flow regimes and transitions

This study shows that the velocity profiles obtained by a cohesive granular medium flowing down an inclined plane are influenced by three main parameters: the cohesion between particles through the potentials of aggregation and fragmentation, the ground friction and the initial velocity of the flow. In this section, we correlate the presented results with the actual four flow regimes of snow avalanches defined by Köhler *et al.* [22]: sliding slab, cold shear, warm shear and plug flow regimes.

The sliding slab regime represents the initiation phase on an avalanche, which normally starts with the release of a dry cohesive snow slab, rapidly fragmenting into smaller blocks along the track [54]. Scenario 5 (Fig 6d–6f) illustrates well this flow regime. Typical experimental values for the tensile strength of slabs of density around 500 kg/m$^3$ are found in the range 10–100 kPa [46, 55]. In this case, we set the bond strength at the higher limit of this range ($\sim$ 100 kPa, $\eta_f = 0.1$) for the sake of sustaining the slab's wholeness as it reaches a steady velocity. We estimate that a scenario with $\eta_f \approx 1$, $\eta_a = 0.1$ and an initial bonding of the particles would come close to a soft snow slab case, which would rapidly fragments in multiple smaller blocks and eventually reach a quasi-cohesionless granular flow. We emphasize that scenario 5 does not correspond to the definition of the plug flow regime given by Köhler *et al.* [22], despite the presence of a vertical velocity profile. Indeed, the velocity of $\sim$ 12 m/s and the brittle behavior of the slab in scenario 5 is inconsistent with the slow, wet snow observed in avalanches plug flow regimes.

The cold dense regime has been defined by Köhler *et al.* [22] as a fast flow (up to around 30 m/s) with a large shearing across the depth. This is what we observe in scenarios 1 (Fig 5a) and 4 (Fig 6a), which both exhibit the same velocity profile despite a very different fragmentation potential. Fig 4 indeed shows that for a low enough aggregation potential, the fragmentation number, and thus the strength of the bonds, do not play any role in the flow dynamics, given that no initial bonding is applied. Thus, we verify that the cold dry snow generating cold dense regimes have a low $\eta_a$. Yet, aggregates are observed even in dry avalanches, remaining from pieces of the released slab [14]. For this reason, the value of $\eta_f$ will directly depend on the strength and thus on the density of the slab with typical values between 0.1 and 1 for hard slabs and higher than 1 for soft slabs. The influence of the ground's roughness on cohesionless flow is shown in S3 Appendix. For the range $0.8 \geq \frac{\bar{r}}{r_g} \geq 4$, we find that decreasing roughness shifts the velocity profile to a higher magnitude.

The warm shear regime is qualitatively similar to the cold dense regime, except that it exhibits lower velocities and shear rates. In the simulations, we observe this effect with an increase of the aggregation potential (Fig 4). Moreover, the effect of fragmentation becomes more important as aggregation increases because more bonds are created. This leads to a decrease of the velocity and the shearing of the flow, as it is the case when comparing scenarios 1 to 2 or 1 to 3. Furthermore, we remark the formation of a plugged layer at the surface while the aggregation number increases (e.g. Fig 5g). This outcome has already been observed by Rognon *et al.* [33] and Brewster *et al.* [53] using a different cohesive contact model. To obtain a warm shear regime, we estimate that the aggregation potential has to be in the range $0.5 < \eta_a < 10$ and $\eta_f$ similar to the previous case, depending on snow density. This regime can then exhibit a wide range of profile shapes, which are set by $\eta_f$ and $\eta_a$.

The plug flow regime is characterized by a velocity below 5 m/s, constant across the depth, as observed in scenario 7 (Fig 8). The conditions necessary to obtain a plug flow were investigated in section *Plug flow regime*. The particles need to form one single "granule", which requires a high $\eta_a$ and low $\eta_f$. Yet, we have observed that the critical property controlling

velocity is not cohesion, unlike the other flow regimes. In contrast, the velocity of the flow is controlled by the ground friction (Fig 7). The initial velocity is not impacting the final one, as long as it is larger than this latter. The roughness of the ground is found to have an effect principally on the effective static friction coefficient (S3 Appendix). With our setup, we find a suitable ground friction of 0.35 to match the observed plug regime, with $\eta_f > 0.5$ and $\eta_a > 5$. The necessity to reduce the bed friction to reproduce reported plug avalanche flows can be interpreted as the lubrication process occurring close to the basal layer in this flow regime, as described by Ancey and Bain [56] and Vera Valero et al. [20]. The presence of meltwater deep in the flow decreases the dry friction [57] and create a sliding surface.

Finally, scenario 2 (Fig 5d–5f) presents a flow that would characterize slush snow. Contrary to the chute flow experiment of Jaedicke et al. [27], we find that the slush flow runs slower than dry snow (Fig 5a). An avalanche with this type of snow has not yet been reported in Vallée de la Sionne but can be relevant in situations such as rain–on–snow events. In this case, bonds break and particles aggregate easily.

The simulated flow regime transition shows some interesting aspects concerning the change in cohesion and ground friction. In Fig 10 for $30 \le t \lesssim 47$ s, we observe that the velocity of the flow is stable, despite important changes of $\eta_f$ and $\eta_a$. This can be explained by the fact that, during this period, the aggregation potential does not yet take over the fragmentation potential. It is also possible to see this effect on the left part of Fig 4 where the velocity profiles are not impacted by a change of $\eta_f$ for low values of $\eta_a$. On Fig 10, the flow field starts to be strongly affected from the moment that $\eta_f$ becomes lower than $\eta_a$. The lowering of the ground friction, required to sustain the plug flow as we have seen previously, imply the formation of a low-frictional layer at the base of the flow during a transitional avalanche.

## Limitations and outlooks

Snow avalanches are geophysical mass flows of high complexity [9, 58] related to the fact that, in natural conditions, snow exists close to its melting point leading to a variety of mechanical behaviors [59, 60]. The challenge of modeling flowing snow is tackled in this work by making various assumptions which are listed below.

The contact model bears two important limitations. The first one is the fact that the bonds between particles are purely elastic–brittle (Fig 2). This does not raise any issue in the case of dry flows where few bonds are present, however, it impacts the dynamics of highly cohesive flows. This statement is well illustrated by scenarios 7 and 5 in the upper layer. In these cases, all the particles are bonded together, which results in larger velocity fluctuations (error bars in Figs 6d and 8). This limitation arises from the fact that almost no viscous dissipation is acting on bonded contacts. The second limitation of the contact model, as stated by Kyburz et al. [12], lies in the fact that it does not allow plastic compaction. As a consequence, density does not change during the flow in contrast to field observations. As important mechanical properties of snow depend on its density (e.g., Young's Modulus [49], tensile strength [61] and shear strength [62]) we think this is a promising direction for improving the model.

Scenario 2 (Fig 5d–5f) presents the flow that would generate a slush snow. However, given the fact that this type of snow is water-saturated, we question the ability of our model to simulate slush. First because of the aforementioned issue of purely elastic bonds that provides a certain rigidity to the flow. Secondly because water saturation greatly decrease the bulk friction of flowing snow, which was kept constant for the simulations presented here. Overall, further experiments would be necessary to better characterize slush flows and more generally to gain a better understanding of the effects of water content and temperature on snow mechanics.

The setup presented in Fig 3 is designed to be as simple as possible in order to investigate flow regimes by only adapting interactions between particles. However, important features are missing for a deeper investigation. First, the constant snow height and slope angle did not permit to study the effect of cohesion on the value of $h_{stop}(\theta)$: the height below which no flow is possible [63]. Then, the fixed snow bed does not allow to explore the mechanisms of erosion, entrainment and deposition which are known to greatly impact avalanche dynamics [54, 64, 65]. Finally, it is worth mentioning that the formation of rounded aggregates, as described for example by Bartelt *et al.* [14] and simulated by Li *et al.* [66] was not modeled here. Fig 6i shows aggregates due to breakage only. In Steinkogler *et al.* [15], successive impacts induced by the tumbler geometry mimics the effect of irregular terrain. Here, while the process of agglomeration is the same, the simple setup geometry prevents from modeling the growth of discrete granules. Therefore, the mechanisms of particle-size segregation [67–69] were not simulated. We argue that simulations using this model over a complex topography would naturally lead to granulation and size-segregation.

All the presented limitations about the setup are naturally possible outlooks for future research, including for example the rheology of snow or the interactions between the various flow regimes and an erodible snow cover.

## Conclusion

In this paper, the Distinct Element Method was used to simulate various flow regimes occurring in snow avalanches. A cohesive contact model was used to mimic solid bonds between snow granules that can form or break depending on the contact force. These aggregation and fragmentation potentials were characterized by two dimensionless numbers which were varied, along with the ground friction, to reproduce different flow regimes identified in full-scale avalanche measurements by Köhler *et al.* [22]. On the one hand, fast sheared flows require low aggregation and high fragmentation. On the other hand, slow plug require high aggregation and low fragmentation propensity as well as a lower ground friction coefficient. The latter potentially reflects basal lubrication in wet snow avalanches. Flow regime transition are successfully simulated by simultaneously and appropriately varying aggregation and fragmentation numbers as well as ground friction. This study thus improves our understanding of important mechanical processes involved in snow avalanche flow regimes and will be extended in the future to explore the effect of temperature on snow rheology and extract constitutive relations with a view of improving depth-averaged avalanche models [70]. The latter is particularly important in a climate change context as global warming affects snow avalanche types with important impacts on hazard mapping procedures.

## Supporting information

**S1 Appendix. Distinct Element Method.** This appendix presents additional information about the DEM: the equations of motion and the equations of force-displacement.
(PDF)

**S2 Appendix. Post processing.** This appendix presents additional information about the post-processing to compute the velocity, stress and pressure.
(PDF)

**S3 Appendix. Sensibility analysis of the ground roughness.** Left: Velocity profiles for the scenario 1. Right: Effective friction of the ground in the case of scenario 7 (plug flow regime).
(PDF)

**S1 Fig. Comparison of the velocity profiles for 2D and 3D simulations for scenario 1.**
(TIF)

## Acknowledgments

The authors acknowledge the WSL's strategic initiative Climate Change Impacts on Alpine Mass Movements (CCAMM) and its members for fruitful discussions that helped to shape this study. The authors also acknowledge the anonymous reviewers for their very constructive feedback.

## Author Contributions

**Conceptualization:** Camille Ligneau, Betty Sovilla, Johan Gaume.

**Data curation:** Camille Ligneau, Betty Sovilla.

**Formal analysis:** Camille Ligneau, Betty Sovilla, Johan Gaume.

**Funding acquisition:** Betty Sovilla.

**Investigation:** Camille Ligneau, Betty Sovilla, Johan Gaume.

**Methodology:** Camille Ligneau, Betty Sovilla, Johan Gaume.

**Project administration:** Camille Ligneau, Betty Sovilla, Johan Gaume.

**Resources:** Camille Ligneau, Betty Sovilla, Johan Gaume.

**Software:** Camille Ligneau, Johan Gaume.

**Supervision:** Camille Ligneau, Betty Sovilla, Johan Gaume.

**Validation:** Camille Ligneau, Betty Sovilla, Johan Gaume.

**Visualization:** Camille Ligneau, Betty Sovilla, Johan Gaume.

**Writing – original draft:** Camille Ligneau, Betty Sovilla, Johan Gaume.

**Writing – review & editing:** Camille Ligneau, Betty Sovilla, Johan Gaume.

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
