## [Decision Letter · Decision Letter 0]

4 Nov 2021

PONE-D-21-31496Effect of cohesion and ground friction on granular flows. Application to snow avalanches.PLOS ONE

Dear Dr. Ligneau,

Thank you for submitting your manuscript to PLOS ONE. After careful consideration, we feel that it has merit but does not fully meet PLOS ONE’s publication criteria as it currently stands. Therefore, we invite you to submit a revised version of the manuscript that addresses the points raised during the review process.

Attached please find the reviewers' comments. Your revision should address the reviewers’ comments as much as you can and include a point-by-point response to the concerns. Please submit your revised manuscript by Dec 19 2021 11:59PM. If you will need more time than this to complete your revisions, please reply to this message or contact the journal office at plosone@plos.org. Please include the following items when submitting your revised manuscript:A rebuttal letter that responds to each point raised by the academic editor and reviewer(s). You should upload this letter as a separate file labeled 'Response to Reviewers'.A marked-up copy of your manuscript that highlights changes made to the original version. You should upload this as a separate file labeled 'Revised Manuscript with Track Changes'.An unmarked version of your revised paper without tracked changes. You should upload this as a separate file labeled 'Manuscript'.

We look forward to receiving your revised manuscript.

Kind regards,

Lei Li

Academic Editor

PLOS ONE

Journal Requirements:

"This research is funded by WSL’s strategic initiative Climate Change Impacts on Alpine Mass Movements (CCAMM) and the canton of Valais. Johan Gaume acknowledges funding from the Swiss National Science Foundation (Eccellenza project: grant number PCEFP2 181227; SPARK project grant number CRSK-2 195914"

"CL and BS are funded by WSL’s  strategic  initiative  Climate  Change  Impacts  on  Alpine  Mass  Movements (CCAMM, ccamm.slf.ch) and the canton of Valais. JG is funded by the Swiss National Science Foundation (Eccellenza project: grant number PCEFP2_181227; SPARK project grant number CRSK-2_195914). The funders had no role in study design, data collection and analysis, decision to publish, or preparation of the manuscript."

Reviewers' comments:

Reviewer's Responses to Questions

**Comments to the Author**

1. Is the manuscript technically sound, and do the data support the conclusions?

Reviewer #1: Yes

Reviewer #2: Yes

Reviewer #3: Partly

2. Has the statistical analysis been performed appropriately and rigorously? 

Reviewer #1: N/A

Reviewer #2: Yes

Reviewer #3: N/A

3. Have the authors made all data underlying the findings in their manuscript fully available?

Reviewer #1: Yes

Reviewer #2: Yes

Reviewer #3: Yes

4. Is the manuscript presented in an intelligible fashion and written in standard English?

Reviewer #1: Yes

Reviewer #2: Yes

Reviewer #3: Yes

5. Review Comments to the Author

Reviewer #1: The article is well written and considers the actual problem of creating mathematical models of snow avalanches, which are a dangerous natural hazard in the mountains. To organize protection, information is needed on the speed of snow movement, impact pressure on the structure and runout distances. Mathematical and numerical modeling is an important tool for obtaining such information. The authors consider an important class of avalanches, which are granular flows. They adopt some simplified model of interaction of granules and numerically investigate the influence of various parameters included in this model, in particular, inter-particle cohesion and ground friction, on snow velocity profiles. The results are compared to data obtained by measurements of some real avalanches.

The article contains very interesting new results. I recommend publishing it.

There are a few minor points that should be clarified.

1. The authors consider 2D flows. Are the particles spherical (line 134) or they are cylinders (line 249)?

2. Please refine the Fig.2 (right) which shows the parallel-bond contact model. What is “a parallel spring-dashpot model for the normal interactions”? It is not a part of the bond since the bond is purely elastic (e.g., line 426). And particles are undeformable (line 134) however their Young modulus and damping ratio are given in Table 1 and shown in Fig.2.

3. The author write “Particles have a mean radius _r = 40 mm with 20% polydispersity” (line 145). However, in calculations, do they study mono-disperse particles (line 196) or not? Please clarify.

Reviewer #2: Dear Authors

From your efforts will be appreciated. However, there are many concerns and shortcomings for your manuscript. Examining the process of snow-flow with such complex and multifaceted physical conditions requires more detailed factors and studies. You must respond carefully to comments and make serious corrections. We need to see how your manuscript is improved in the next step.

Reviewer #3: In the manuscript, the authors performed and analysed a series of DEM simulations with PFC v5 (Itasca). The main purpose is to reproduce different avalanche regimes by changing the cohesion (that can be described using two dimensionless parameters and ) and the ground friction (described by the friction parameters ). Two simulations are then compared to a real avalanche that happened in the test site Vallée de la Sionne on March 8th 2017 where two distinct flow regimes are highlighted. Finally, a simulation of the entire event is analysed where the parameters are linearly changed.

All the observation are listed in the attached file Revision.pdf

6. PLOS authors have the option to publish the peer review history of their article (what does this mean?). If published, this will include your full peer review and any attached files.

Reviewer #1: No

Reviewer #2: No

Reviewer #3: No

---

## [Author Response · Author response to Decision Letter 0]

29 Dec 2021

Please see the attached file 'Response to Reviewers'.

---

## [Decision Letter · Decision Letter 1]

21 Jan 2022

PONE-D-21-31496R1Numerical investigation of the effect of cohesion and ground friction on snow avalanches flow regimesPLOS ONE

Dear Dr. Ligneau,

Thank you for submitting your manuscript to PLOS ONE. After careful consideration, we feel that it has merit but does not fully meet PLOS ONE’s publication criteria as it currently stands. Therefore, we invite you to submit a revised version of the manuscript that addresses the points raised during the review process.

Before the paper is accepted, please address the minor revisions requested by the reviewer 3.

We look forward to receiving your revised manuscript.

Kind regards,

Lei Li

Academic Editor

PLOS ONE

Journal Requirements:

Reviewers' comments:

Reviewer's Responses to Questions

**Comments to the Author**

1. If the authors have adequately addressed your comments raised in a previous round of review and you feel that this manuscript is now acceptable for publication, you may indicate that here to bypass the “Comments to the Author” section, enter your conflict of interest statement in the “Confidential to Editor” section, and submit your "Accept" recommendation.

Reviewer #1: All comments have been addressed

Reviewer #3: All comments have been addressed

2. Is the manuscript technically sound, and do the data support the conclusions?

Reviewer #1: Yes

Reviewer #3: Yes

3. Has the statistical analysis been performed appropriately and rigorously? 

Reviewer #1: I Don't Know

Reviewer #3: N/A

4. Have the authors made all data underlying the findings in their manuscript fully available?

Reviewer #1: Yes

Reviewer #3: Yes

5. Is the manuscript presented in an intelligible fashion and written in standard English?

Reviewer #1: Yes

Reviewer #3: Yes

6. Review Comments to the Author

Reviewer #1: (No Response)

Reviewer #3: The authors gave convincing answers to all my observations and significantly improved their manuscript. It remains only some minor observations that can be founded in the attachment.

7. PLOS authors have the option to publish the peer review history of their article (what does this mean?). If published, this will include your full peer review and any attached files.

Reviewer #1: No

Reviewer #3: No

---

## [Author Response · Author response to Decision Letter 1]

28 Jan 2022

See attached file "Response to Reviewers"

---

## [Editor Report · Decision Letter 2]

2 Feb 2022

Numerical investigation of the effect of cohesion and ground friction on snow avalanches flow regimes

PONE-D-21-31496R2

Dear Dr. Ligneau,

We’re pleased to inform you that your manuscript has been judged scientifically suitable for publication and will be formally accepted for publication once it meets all outstanding technical requirements.

Kind regards,

Lei Li

Academic Editor

PLOS ONE
---

## [Editor Report · Acceptance letter]

7 Feb 2022

PONE-D-21-31496R2 

Numerical investigation of the effect of cohesion and ground
friction on snow avalanches flow regimes. 

Dear Dr. Ligneau:

I'm pleased to inform you that your manuscript has been deemed suitable for publication in PLOS ONE. Congratulations! Your manuscript is now with our production department. 

Kind regards, 

on behalf of

Dr. Lei Li 

Academic Editor

PLOS ONE